# Recruitment of a splicing factor to the nuclear lamina for its inactivation

Karen Vester [1✉], Marco Preußner[2], Nicole Holton[1], Suihan Feng [3], Carsten Schultz[3], Florian Heyd[2] & Markus C. Wahl [1,4✉]

Precursor messenger RNA splicing is a highly regulated process, mediated by a complex RNA-protein machinery, the spliceosome, that encompasses several hundred proteins and five small nuclear RNAs in humans. Emerging evidence suggests that the spatial organization of splicing factors and their spatio-temporal dynamics participate in the regulation of splicing. So far, methods to manipulate the spatial distribution of splicing factors in a temporally defined manner in living cells are missing. Here, we describe such an approach that takes advantage of a reversible chemical dimerizer, and outline the requirements for efficient, reversible re-localization of splicing factors to selected sub-nuclear compartments. In a proof-of-principle study, the partial re-localization of the PRPF38A protein to the nuclear lamina in HEK293T cells induced a moderate increase in intron retention. Our approach allows fast and reversible re-localization of splicing factors, has few side effects and can be applied to many splicing factors by fusion of a protein tag through genome engineering. Apart from the systematic analysis of the spatio-temporal aspects of splicing regulation, the approach has a large potential for the fast induction and reversal of splicing switches and can reveal mechanisms of splicing regulation in native nuclear environments.

[1] Freie Universität Berlin, Institute of Chemistry and Biochemistry, Laboratory of Structural Biochemistry, Takustrasse 6, D-14195 Berlin, Germany. [2] Freie Universität Berlin, Institute of Chemistry and Biochemistry, Laboratory of RNA Biochemistry, Takustrasse 6, D-14195 Berlin, Germany. [3] Oregon Health and Science University, Department of Chemical Physiology and Biochemistry, 3181 SW Sam Jackson Park Rd., L334, Portland, OR 97239, USA. [4] Helmholtz-Zentrum Berlin für Materialien und Energie, Macromolecular Crystallography, Albert-Einstein-Straße 15, D-12489 Berlin, Germany. ✉email: karenvester@zedat.fu-berlin.de; markus.wahl@fu-berlin.de

Precursor messenger RNA (pre-mRNA) splicing entails the removal of non-coding regions (introns) from pre-mRNAs and the concomitant ligation of neighboring coding regions (exons). Most human pre-mRNAs contain more than one intron and can be spliced in alternative ways to give rise to multiple mature mRNAs and thus multiple proteins[1]. Pre-mRNA splicing is mediated by the spliceosome, an RNA-protein (RNP) molecular machine of stunning complexity[2,3]. For each round of splicing, a spliceosome is assembled de novo on a substrate pre-mRNA in a stepwise fashion; after extensive remodeling, an active spliceosome is obtained that facilitates the two transesterification steps of a splicing reaction and is subsequently disassembled in an ordered fashion[2]. Alternative splicing is intricately intertwined with the dynamic spliceosome assembly and is regulated on multiple levels, including the sub-nuclear localization of splicing factors[4–7].

Splicing factors predominantly accumulate in Cajal bodies and nuclear speckles, membrane-less compartments that likely form by phase separation[8]. Despite the high concentration of splicing factors, Cajal bodies and speckles are not the sites where splicing predominantly takes place. Cajal bodies constitute sites of small nuclear (sn) RNP biogenesis, where the assembly of the U4/U6 di-snRNP and the U4/U6-U5 tri-snRNP, the final maturation steps of the U2 snRNP as well as 2′-O-methylation and pseudo-uridylation of snRNAs have been shown to occur[9–11]. Furthermore, the tri-snRNP is re-assembled in Cajal bodies after its major remodeling during each round of splicing[9]. Speckles are largely devoid of nascent RNAs[12,13] and are thought to represent storage sites for splicing factors. When genes are transcribed, chromatin often forms loops that reach out into the peri-chromatin region[14], i.e., the region between chromatin and the inter-chromatin space[15]. Splicing is thought to predominantly take place co-transcriptionally[16] and in peri-chromatin fibrils[17,18], where splicing factors can be delivered from nuclear speckles and where they co-localize with nascent transcripts in an intron-dependent manner[19].

The potential functional interplay between splicing and nuclear compartmentalization has been analyzed by imaging cell sections, e.g., via electron microscopy, fluorescence in situ hybridization, 5-bromouridine-5′-triphosphate labeling, and optical microscopy techniques[19–21]. For instance, the exchange of factors between speckles and the nucleoplasm is implicated by speckle extensions and factor dissociation at the periphery of speckles[22]. Furthermore, upon inhibition of transcription and splicing, the appearance of speckles changes, with a lower degree of peripheral dynamics and an increase in speckle size, which likely reflects the enhanced storage of factors in nuclear speckles when spliceosome assembly is not required[22–24]. In addition, upon treatment of cells with transcription inhibitors, the nuclear distribution of splicing factors or their co-localization with other factors have been observed to change[22,25].

Although the above findings suggest that the intricate compartmentalization of the nucleus into nuclear bodies, chromatin territories, inter-chromatin space, and peri-chromatin regions most likely plays a major role in the regulation of splicing, the consequences of the localization of individual splicing factors remain to be elucidated. To this end, the development of methods that allow the precise re-localization of splicing factors to selected subcellular or sub-nuclear compartments in a temporally controlled manner is required.

Here, we present an approach that allows the fast and reversible recruitment of splicing factors to selected nuclear regions via reversible chemically induced dimerization (rCID). Chemical dimerizers are compounds that concomitantly bind to two protein domains. Upon fusion of these domains to two target proteins, the compounds can induce hetero-dimerization of these targets. These tools have been used for the analysis of cellular signaling pathways, for example, G-proteins[26] and enzymes involved in lipid metabolism[27]. We employed a recently developed reversible chemical dimerizer, rCD1, that mediates the interaction between a SNAP-tagged protein and an FKBP-tagged protein[28]. The chemical dimerization can be reversed by the addition of FK506, which efficiently competes with the rCD1-FKBP interaction. We show that fusion of the FKBP domain to a target splicing factor and concomitant fusion of the SNAP-tag to an anchor protein that exhibits a defined sub-nuclear localization allows for the recruitment of the splicing factor to the respective sub-nuclear region. We conducted a case study with the PRPF38A protein as the target splicing factor that plays an important role during spliceosome activation. rCID-based recruitment of PRPF38A to the nuclear lamina induced a mild increase in intron retention according to RNA sequencing results and radioactive PCR analyses. Our approach can be adapted to other splicing factors and anchor proteins to elucidate the effects of re-localizing individual splicing factors to selected sub-nuclear compartments in a temporally defined manner.

## Results

**Identification of a suitable anchor protein for rCID-mediated recruitment.** Recruitment of spliceosomal target proteins to anchor proteins that reside in subcellular regions that do not support active pre-mRNA splicing might cause splicing deficiencies or even abolish splicing. To test this hypothesis, we first tested different SNAP-tagged anchor proteins for the recruitment of FKBP-tagged splicing factor, PRPF38A. PRPF38A was identified as an essential splicing factor in yeast and was found to be important for the progression of the splicing process to the first splicing step in this organism[29]. PRPF38A-depleted yeast extract exhibits a defect in U4 snRNA release and spliceosome activation[30]. In humans, PRPF38A is one of nine splicing factors that are specifically incorporated during the formation of the pre-catalytic spliceosomal B complex (B-specific proteins) and that are released again during spliceosome activation[31,32]. Additionally, PRPF38A is involved in a multitude of protein–protein interactions, such as with the B-specific proteins MFAP and Snu23[33–36]. Due to its transient incorporation into the spliceosome, we considered PRPF38A as a promising target for the rCID-based re-localization. The addition of rCD1 induces dimerization of a FKBP-tagged protein and a SNAP-tagged protein (Fig. 1a). We, thus, monitored the cellular localizations of transiently expressed anchor-ECFP-SNAP and RFP-FKBP-PRPF38A fusions via the fluorescent ECFP/RFP moieties in the absence and presence of rCD1 by using confocal microscopy. All anchor constructs showed the expected sizes as ECFP-SNAP fusion proteins according to a Western blot (Fig. S1).

As a plasma membrane (PM)-localized anchor, we chose a PM-targeting peptide sequence of the kinase Lck, which has previously been employed to recruit the phosphoinositide 5-phosphatase to the PM[37]. The addition of the dimerizer to HEK293T cells transfected with vectors expressing Lck-ECFP-SNAP and RFP-FKBP-PRPF38A proteins did not instigate recruitment of the PRPF38A fusion construct to the PM (Fig. 1b). Even incubation of the cells with the dimerizer for 24 h, during which they underwent cell division and thus the intermittent disruption of the nuclear envelope, was not sufficient to induce translocation (Supplementary Fig. S2a). The CAAX motif targets proteins to the endoplasmic reticulum (ER) and Golgi and, ultimately, via the endomembrane system to the PM[38]. However, we likewise did not observe a co-localization of RFP-FKBP-PRPF38A with ECFP-SNAP-CAAX at the endomembrane system or the PM after the addition of rCD1 (Fig. 1c and Supplementary Fig. S2b).

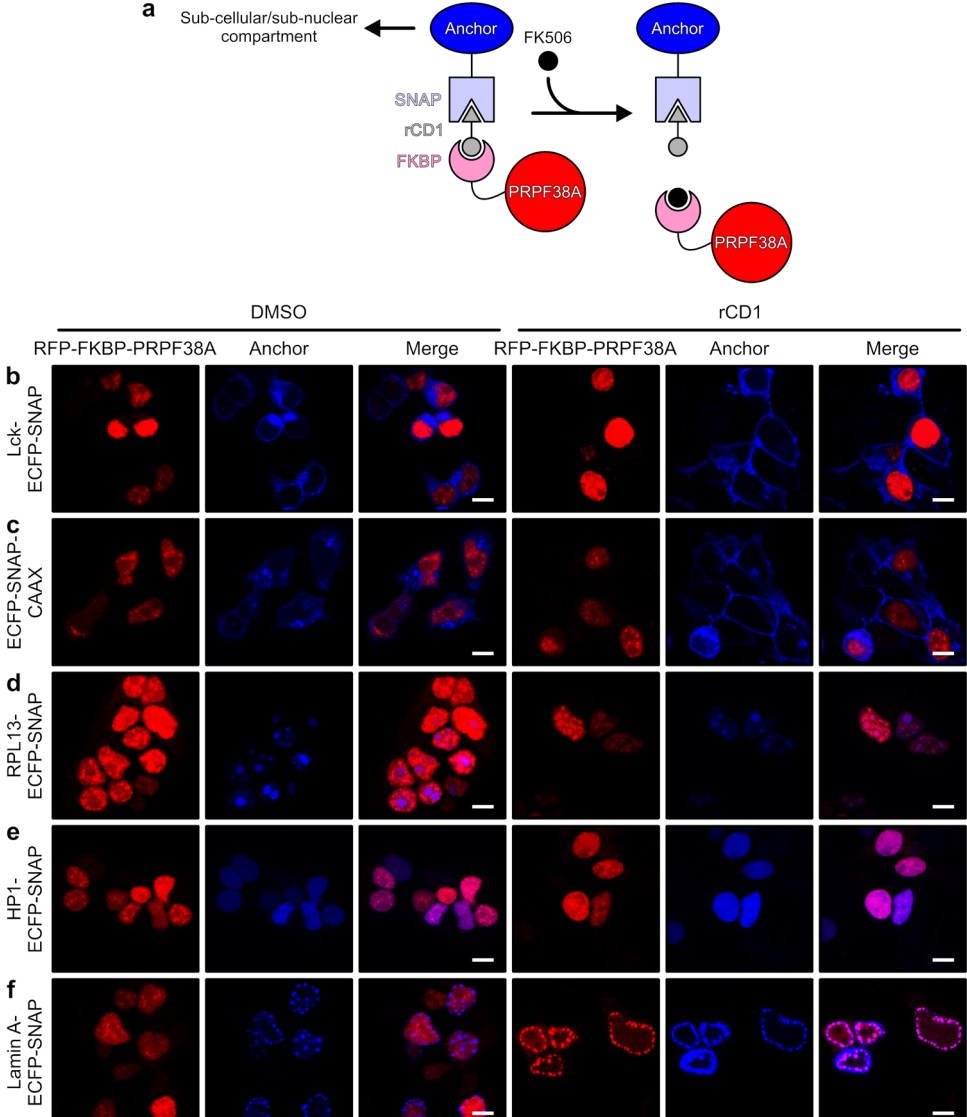

**Fig. 1 Test of anchor proteins. a** Dimerization principle of rCD1. The compound rCD1 induces dimerization of a SNAP-tag and an FKBP tag. The FKBP-tagged target protein is recruited to the SNAP-tagged anchor protein. The dimerization is reversed upon the addition of FK506. **b–e** HEK293T cells were co-transfected with vectors expressing the indicated (on the left) anchor-ECFP-SNAP or ECPF-SNAP-anchor protein fusions (blue channel) and the target-RFP-FKBP-PRPF38A protein fusion (red channel). The localization of the proteins was monitored in DMSO-treated cells (groups of three panels on the left) or after rCD1 treatment (groups of three panels on the right) for 30 min. Each group of three images shows the red channel (RFP-FKBP-PRPF38A; left), the blue channel (anchor-ECFP-SNAP or ECFP-SNAP-anchor; middle), and a merged image (right), as indicated on the top. Scale bars (10 μm) are shown as white lines on the bottom right of the merged images. The images did not indicate successful recruitment of RFP-FKBP-PRPF38A to **b** Lck-ECFP-SNAP at the PM, **c** ECFP-SNAP-CAAX at the endomembrane system and the PM, and **e** HP1-ECFP-SNAP in heterochromatin regions. In contrast, partial recruitment to **d** RPL13-ECFP-SNAP in nucleoli and recruitment to **f** lamin A-ECFP-SNAP at the lamina upon rCD1 addition could be detected.

In the following tests, we, therefore, used nuclear proteins as anchors, which were expected to exhibit a higher probability of encountering nuclear-localized splicing factors. The over-expressed ribosomal protein fusion, RPL13-ECFP-SNAP, showed a nucleolar localization (Fig. 1d). Although ribosomal proteins are usually localized to the cytoplasm, their over-expression can lead to nucleolar accumulation, because the proteins are in excess to other ribosomal components and, thus, not assembled into ribosomal particles and not exported into the cytoplasm[39,40]. After the addition of the dimerizer, we observed an increase in co-localization of the PRPF38A fusion construct with RPL13-ECFP-SNAP in nucleoli (Fig. 1d). Whereas the localization of RPL13-ECFP-SNAP was unchanged during the course of the treatment, RFP-FKBP-PRPF38A was recruited to

nucleoli within 30 min (Fig. 1d). However, the recruitment was not complete, as residual RFP-FKBP-PRPF38A was still localized to the nucleoplasm (Fig. 1d).

As further anchors, we considered proteins with a localization distinct from euchromatin. In this way, we expected the spliceosomal target proteins to be sequestered from the euchromatin-associated peri-chromatin fibrils, the hot spots of splicing activity. Heterochromatin protein 1 (HP1), a major component of heterochromatin, fused to ECFP-SNAP exhibited a high degree of co-localization with RFP-FKBP-PRPF38A already before the addition of rCD1, so the addition of the dimerizer only induced a slight increase in co-localization (Fig. 1e). One reason could be a partial localization of HP1 to euchromatin, as has previously been described[41]. In addition,

the distribution of HP1 between hetero- and euchromatin might be influenced by overexpression.

Finally, we tested the dimerizer system for the recruitment of spliceosomal proteins to the nuclear lamina. The lamina is located below the inner nuclear membrane and is associated with certain chromatin regions, so-called lamina-associated domains[42]. These domains were shown to be heterochromatic regions that are often associated with low gene expression levels[43]. As an anchor protein, we selected the protein lamin A, which is predominantly localized at the lamina but also has a nucleoplasmic pool[44]. To achieve exclusive localization of lamin A at the lamina, we attached the tags to the C-terminus of the protein, which abolished the C-terminal processing of lamin A, resulting in a depletion of the nucleoplasmic lamin A pool[45,46]. As expected, the lamin A-ECFP-SNAP construct was only localized at the lamina, accompanied by the formation of spherical aggregates (Fig. 1f). The tendency for aggregation was dependent on the level of overexpression of the lamin A fusion, as cells with lower transfection levels showed lower levels of spherical aggregates. The addition of rCD1 resulted in an efficient recruitment of RFP-FKBP-PRPF38A to lamin A-ECFP-SNAP at the lamina (Fig. 1f). Therefore, we focused on this combination in further experiments.

**Recruitment kinetics and reversal**. Recruitment kinetics determine the temporal precision of the system. To characterize recruitment kinetics, we conducted time-dependent live-cell confocal microscopy to monitor the recruitment within the same cells over time. Imaging after rCD1 addition was started 2 min after treatment as the earliest possible time point due to sample handling. The serial images showed increasing co-localization of RFP-FKBP-PRPF38A to lamin A-ECFP-SNAP at the lamina over time (Fig. 2a). Quantifying the degree of co-localization revealed a clear time-dependent increase in co-localization with saturation after 30–40 min (Fig. 2b). The observed recruitment kinetics are thus very similar to those observed for rCD1-based recruitment of RFP-FKBP to the PM via Lck-ECFP-SNAP, which also reached saturation after approximately 40 min[28]. The requirement of rCD1 to diffuse through several layers of membrane barriers in our experiments, therefore, does not seem to represent a rate-limiting step for the recruitment. Furthermore, rCD1-mediated dimerization persisted at least for an entire day without an obvious decrease in the level of co-localization.

rCD1 has been designed to allow rapid reversal of dimerization upon addition of FK506, which binds to FKBP and outcompetes rCD1[28]. Indeed, RFP-FKBP-PRPF38A recruitment to lamin A-ECFP-SNAP at the lamina was completely reversed within 5 min after the addition of FK506 to cells that had previously been exposed to rCD1 and exhibited maximum co-localization (Fig. 2c). Similarly, RPL13-ECFP-SNAP-mediated recruitment of RFP-FKBP-PRPF38A to nucleoli occurred within 30 min (Fig. S3a) and was quickly reversed by addition of FK506 (Fig. S3b). FK506-induced reversal was too fast for accurate quantification of the kinetics. In conclusion, the rCD1-based recruitment takes effect rapidly within 30–40 min after rCD1 addition and can be reversed by FK506 in less than 5 min.

**Anchoring of other spliceosomal proteins**. To test if the identified anchor proteins can also be used to recruit other spliceosomal proteins, we tested the anchoring of U2AF35. U2AF35 interacts with U2AF65 to form the essential U2AF splicing factor complex. U2AF35 interacts with the AG dinucleotide of the 3′-splice site[47], and the U2AF complex subsequently assists in the incorporation of U2 snRNP into the spliceosomal A complex[48]. U2AF35 and the U2AF complex are required for constitutive splicing and are involved in the regulation of alternative splicing[49–52].

As for PRPF38A, we co-expressed anchor-ECFP-SNAP and RFP-FKBP-U2AF35 in HEK293T cells. In contrast to RFP-FKBP-PRPF38A, which accumulated in nuclear speckles, RFP-FKBP-U2AF35 was rather evenly distributed in the nucleoplasm before the addition of rCD1 (Fig. 3), as has been also observed for the U2AF complex[53]. Addition of rCD1 induced recruitment of RFP-FKBP-U2AF35 to RPL13-ECFP-SNAP in nucleoli and to lamin A-ECFP-SNAP at the lamina within 30 minutes (Fig. 3), suggesting that rCD1-mediated anchoring can be applied to diverse splicing factors.

**On-locus integration of an FKBP-coding region via genome engineering**. Individual cells showed different efficiencies in the recruitment of RFP-FKBP-PRPF38A to lamin A-ECFP-SNAP at the lamina (Fig. 2d), dependent on the ratio of the target and anchor protein fusions. For cells with a high relative expression level of the lamin A construct, the recruitment worked efficiently, while for cells with a relatively low expression level of the lamin A construct, the recruitment was incomplete, as indicated by higher red fluorescence remaining in the nucleoplasm (Fig. 2d). Therefore, with higher transfection and expression efficiency of the anchor construct, a more efficient recruitment can be achieved. Additionally, co-transfection was not complete, and recruitment cannot be induced in cells that express only one of the two fusion constructs (Fig. 2d).

As co-transfection always results in inhomogeneous cell populations, we decided to fuse an FKBP tag-encoding region to the endogenous *prpf38A* gene in HEK293T cells by CRISPR/Cas9-based genome engineering. We designed four gRNAs targeting the region around the *prpf38A* start codon with good predicted efficiencies and specificities (Supplementary Table S2). Based on a cleavage efficiency test (Supplementary Fig. S4), we used gRNA3 together with a repair template containing an FKBP-coding region for homology-directed repair-based knock-in. Approximately 50 colonies were screened, of which three showed a homozygous integration (Fig. 4a–c). Sequencing of the PCR-amplified genomic DNA confirmed the correct integration in two of the three colonies (cell lines C1 and C2), whereas one showed sequence errors and was therefore not used further.

Despite the homozygous integration on the DNA level, a Western blot with an anti-PRPF38A antibody revealed two prominent bands for both engineered cell lines, one with the expected size of FKBP-tagged PRPF38A (50 kDa) and one with a size of 38 kDa, likely corresponding to PRPF38A without tag (Fig. 4d, left). We tested whether the anti-PRPF38A antibody recognizes an additional protein with a molecular mass of 38 kDa by downregulating the CRISPR-derived FKBP-PRPF38A fusion with a siRNA (in a doubly engineered FKBP-PRPF38A CRISPR/lamin A-ECFP-SNAP stable cell line; see below). The siRNA reduced both the 50 kDa and the 38 kDa bands in Western blots (Fig. 4e), confirming the specificity of the antibody.

To test whether the formation of untagged PRPF38A occurred due to translation re-initiation at the ATG start codon of the *prpf38A* region, which was retained in the first round of genome engineering, we again applied CRISPR/Cas9 to exchange this ATG codon to a CTG codon, which would guide the incorporation of a leucine instead of a methionine residue. Despite the successful homozygous on-site mutagenesis, Western blotting revealed an even higher relative amount of tag-less PRPF38A compared to FKBP-PRPF38A (Fig. 4d, right), excluding translation re-initiation at the authentic *prpf38A* start-ATG as a source of tag-less PRPF38A.

Altered splicing events after CRISPR editing have been described in several studies[54–56]. However, we did not notice aberrant *prpf38A* splicing patterns in RNA sequencing (RNAseq)

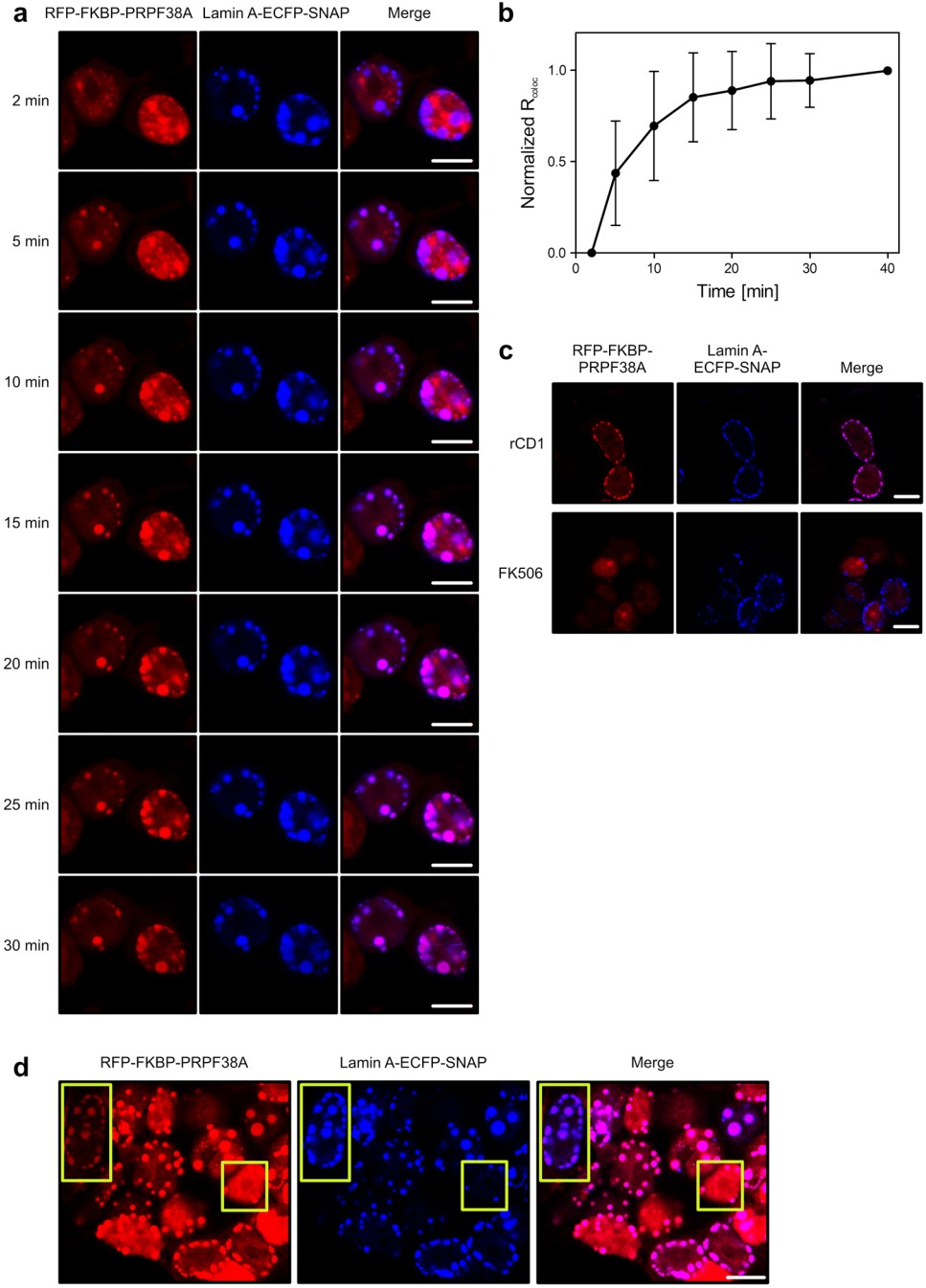

analyses (see below) that could explain the production of tag-less PRPF38A in the engineered cell lines. We further characterized the CRISPR knock-ins on the mRNA level. Radioactive PCR amplification with primers binding within the 5′-untranslated region (UTR) and at the 3′-end of *prpf38A* exon1 (E1; Supplementary Table S1) showed the expected size of *prpf38A* mRNA for the wild-type (WT) HEK293T cells and an additional prominent band, which might belong to another *prpf38A* isoform (Fig. 4f). The CRISPR knock-in cell lines lacked the WT *prpf38A* bands and instead gave rise to bands with the expected size for an *fkbp-prpf38A* fusion mRNA (Fig. 4f). We sequenced the PCR-amplified products of the CRISPR cell lines, which confirmed the correct integration of the FKBP-coding region on the *prpf38A* locus.

Based on the above analyses, we conclude that the properly installed FKBP tag is in part removed by a posttranscriptional

process. One possibility is that the engineered cell lines produce FKBP-PRPF38A fusion protein, but that the linker region between the FKBP tag and PRPF38A is partially cleaved by an endogenous protease. Consistent with this explanation, a cryogenic electron microscopy (cryoEM) structure of a yeast B complex spliceosome[57] suggests that the 20 N-terminal amino acid residues of PRPF38A are unstructured (Supplementary Fig. S5), possibly granting access to a cellular protease. Another possibility is translation re-initiation at an ATG codon in the proximity of the authentic *prpf38A* start-ATG. Such mechanisms have been described before for CRISPR-edited cell lines[56].

**Stable expression of an anchor protein fusion**. Next, we additionally integrated a coding region for the anchor protein fusion into the genome of the FKBP-PRPF38A CRISPR cell line C1. To

**Fig. 2 Characterization of the recruitment kinetics. a** Time-dependent, rCD1-mediated recruitment of the RFP-FKBP-PRPF38A target protein fusion (red channel; left) to the lamin A-ECFP-SNAP anchor protein fusion (blue channel; middle), monitored via live-cell confocal microscopy. Times on the left are after the addition of rCD1. Right panels, merged images. Scale bars (10 μm) are shown as white lines on the bottom right of the merged images.
**b** Quantification of the recruitment kinetics. Nuclei of cells expressing both RFP-FKBP-PRPF38A target and lamin A-ECFP-SNAP anchor protein fusions were selected as regions of interest (ROIs), and the degree of co-localization was quantified via $R_{coloc}$ (Pearson's correlation coefficient for pixels of the two channels). $R_{coloc}$ values for individual ROIs were normalized by subtraction of the $R_{coloc}$ value of the respective ROI for the first time point after rCD1 addition (2 min) and division by the corresponding $R_{coloc}$ value for the respective ROI 40 min after rCD1 addition. Only cells that produced both constructs were used for quantification as ROIs. Data represent means ± SD of 19 quantified ROIs of two monitored treatment responses. **c** Reversal of rCD1-mediated RFP-FKBP-PRPF38A recruitment to lamin A-ECFP-SNAP (upper panels) upon addition of FK506 (lower panels). Cells as in **a** treated with rCD1 for 30 min (upper panels) were additionally incubated with FK506 for 5 min (lower panels). Left panels, red channel showing localization of RFP-FKBP-PRPF38A; middle panels, blue channel showing localization of lamin A-ECFP-SNAP; right panels, merged images. Scale bars (10 μm) are shown as white lines on the bottom right of the merged images. Please note that another part of the same uncropped raw image of rCD1-treated cells (upper panels) was also used for images shown in Fig. 1f. **d** Variable recruitment efficiencies in individual cells. HEK293T cells co-transfected with vectors expressing the RFP-FKBP-PRPF38A target protein fusion (red channel) and the lamin A-ECFP-SNAP anchor protein fusion (blue channel) show different expression ratios of the fusion proteins, which lead to different recruitment efficiencies in different cells after rCD1 treatment for 60 min. The yellow boxes highlight a cell with a low expression level of RFP-FKBP-PRPF38A compared to lamin A-ECFP-SNAP and therefore efficient recruitment (left yellow box) and a cell with a high expression level of the target versus anchor construct and therefore less efficient recruitment (right yellow box). Right panel, merged images. Scale bars (10 μm) are shown as white lines on the bottom right of the merged images.

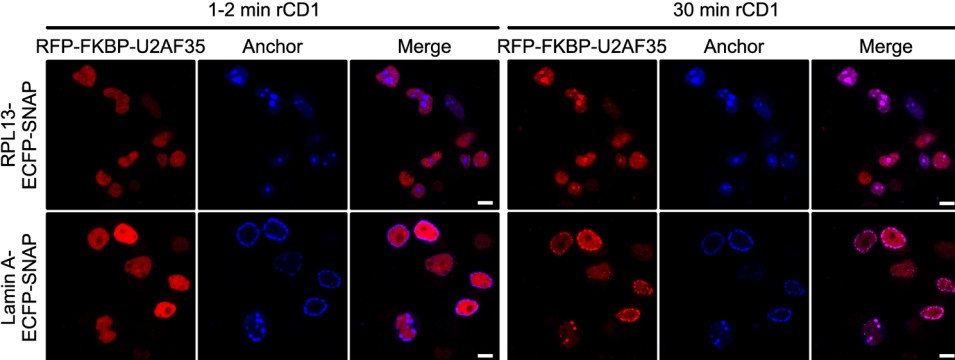

**Fig. 3 rCID-mediated re-localization of U2AF35.** Time-dependent, rCD1-mediated recruitment of the RFP-FKBP-U2AF35 target protein fusion (red channel; left) to the RPL13-ECFP-SNAP (top panels) or lamin A-ECFP-SNAP (bottom panels) anchor protein fusion (blue channels; middle), monitored via live-cell confocal microscopy. Right panels in each group, merged images. The same cells were imaged 1–2 min (left groups of three images) and 30 min (right groups of three images) after the addition of rCD1. Scale bars (10 μm) are shown as white lines on the bottom right of the merged images.

obtain a higher expression level of the anchor protein fusion over the target protein fusion, we stably integrated the lamin A-ECFP-SNAP-coding region into the genome of the C1 CRISPR cell line. Stable integration usually results in the integration of several copies of the insert and our vector contained a strong CMV promoter, so a high expression level of the anchor protein fusion after stable integration into the genome was expected. In contrast, the target FKBP-PRPF38A is present only at endogenous levels after tag knock-in. Stable integration of the anchor lamin A-ECFP-SNAP construct was validated on the DNA level by PCR analysis, revealing the presence of regions encoding ECFP and SNAP (Fig. 5a, b). Additionally, the stable cell line showed the expected ECFP fluorescence (Fig. 5c). We observed that the fluorescence intensities were not identical in all cells, most likely due to cells in different phases of the cell cycle; for example, lamina meshworks, including polymerized lamins, are disassembled during mitosis[58]. The homozygous CRISPR integration of the FKBP-coding region to the *prpf38a* locus was again verified after stable integration of the lamin A-ECFP-SNAP-coding region (Fig. 5a, b).

To test the recruitment of FKBP-PRPF38A in our CRISPR cell line that also stably expressed the lamin A-ECFP-SNAP fusion protein (doubly engineered, DE cells), we treated the cells with rCD1 for 24 h. After fixation and permeabilization, staining was performed with an anti-PRPF38A primary and an Alexa488-coupled secondary antibody. We then recorded Z-stacks through

the untreated or rCD1-treated cells. Alexa488 and ECFP signals co-localized only after rCD1 treatment (Fig. 5d), confirming efficient recruitment of FKBP-PRPF38A to the lamina by rCID. Complete recruitment of all cellular PRPF38A was not achieved due to the presence of the untagged pool of PRPF38A still remaining in the doubly engineered FKBP-PRPF38A CRISPR/lamin A-ECFP-SNAP stable cell line. We suggest that this untagged pool of PRPF38A gives rise to the remaining antibody staining of the nucleoplasm after rCD1 treatment (Fig. 5d).

**Effect of FKBP-PRPF38A recruitment to the lamina on pre-mRNA splicing.** To assess the effect of the recruitment of cellular PRPF38A to the lamina on pre-mRNA splicing, we analyzed the transcriptome of DMSO-treated and rCD1-treated doubly engineered FKBP-PRPF38A CRISPR/lamin A-ECFP-SNAP stable cells by RNAseq[59]. Furthermore, the FKBP-PRPF38A CRISPR singly engineered cell line without stably integrated lamin A-ECFP-SNAP-coding region was treated with DMSO or rCD1 and subjected to transcriptome sequencing to monitor the effects of the treatment independent of the recruitment. The sequencing results were evaluated for effects on alternative splicing with the RMATS tool and for effects on intron retention with the iREAD tool.

rCD1 treatment for 24 h had only minor effects on alternative splicing events. We observed only sporadic changes in exon skipping events, while alternative usage of 3′-splice sites or 5′-splice sites and mutually exclusive exons were not globally influenced by

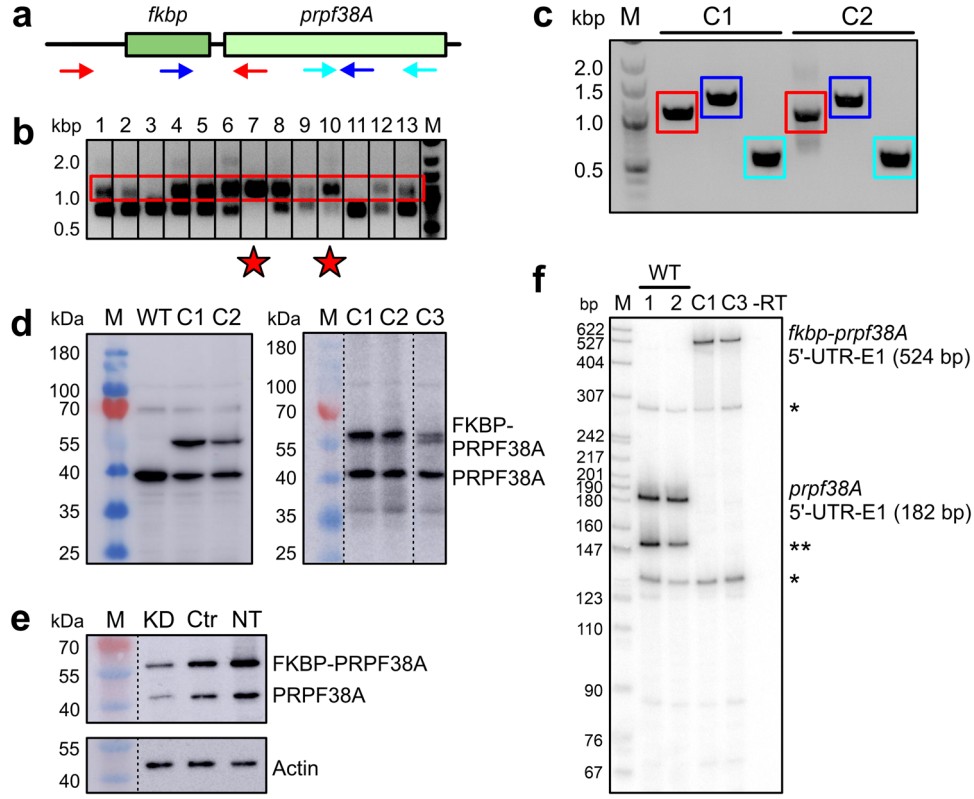

**Fig. 4 CRISPR/Cas9 integration of an FKBP-coding region into the *prpf38A* locus. a** Scheme depicting primer pairs (arrows; Supplementary Table S1) used in experiments shown in **b**, **c**. **b** CRISPR/Cas9-mediated integration of an FKBP-coding region upstream of the PRPF38A-coding region on the endogenous *prpf38A* locus. Engineered clones (a subset of 50 clones shown in lanes 1–13) were tested by PCR with the red primer pair in **a**. Amplified bands originating from successful *fkpb* insertion are boxed in the same color as the respective primer pair in **a**. Heterozygously edited clones show two bands, homozygously edited clones exhibit a single upper band. Star symbols, homozygously edited clones. M, size marker (kbp). **c** Homozygous *fkbp* integration in clones C1 and C2 was further validated by PCR analyses and sequencing of the amplified products. Amplified bands are boxed in the same color as the respective primer pairs in **a**. **d** Left panel, Western blot analysis with an anti-PRPF38A antibody of cell cultures from WT and the two homozygously edited clones C1 and C2. Right panel, Western blot analysis with an anti-PRPF38A antibody of the CRISPR cell lines C1 and C2 in comparison to the cell line C3, in which the authentic start-ATG (methionine) of the *prpf38A*-coding region was additionally converted to a CTG (leucine) codon by CRISPR/Cas9-based engineering. The lanes of the right panel originated from the same gel but were rearranged for display purposes as indicated by dashed lines. **e** Western blot analysis with an anti-PRPF38A antibody after siRNA-mediated knockdown of FKBP-PRPF38A in doubly engineered FKBP-PRPF38A CRISPR/lamin A-ECFP-SNAP stable cell line. Cells were analyzed two days after treatment with *prpf38A* siRNA (KD), control siRNA (Ctr), or no treatment (NT). Bottom panel, staining of the stripped membrane with an anti-actin antibody as a loading control. **f** Radioactive PCR with a forward primer against the 5′-UTR and a reverse primer at the 3′-end of the first exon (E1) of the *prpf38A* gene. Cell lines analyzed are indicated on the top. WT1/2, two independently prepared samples from WT HEK293T cells; C1, original homozygously edited CRISPR cell line C1; C3, doubly CRISPR-edited cell line C3; -RT, PCR analysis without the addition of reverse transcriptase upon cDNA generation (control for the background generated from undigested genomic DNA). *non-specifically amplified products based on sequencing results; **possible alternative *prpf38A* splice variant in WT cells. M, size marker (bp).

the treatment. However, evaluation of the sequencing results for constitutive splicing indicated significant changes in intron retention upon rCD1 treatment of the doubly engineered cell line compared to all three controls (Fig. 6a). In contrast, intron retention of the affected pre-mRNAs was not significantly altered by rCD1 treatment in the FKBP-PRPF38A CRISPR cell line lacking lamin A-ECFP-SNAP integration (Fig. 6a). This hints to increased intron retention due to the recruitment to the anchor in DE cells. Overall, we observed 197 significantly altered intron retention events in rCD1-treated doubly engineered cells compared to all three controls ($p < 0.001$; 163 events [about 83%] with increased intron retention; 34 events [about 17%] with decreased intron retention; examples in Fig. 6b, c).

We validated selected intron retention events altered specifically in the doubly engineered cell line upon rCD1 treatment. PCR reactions were performed with radioactively labeled primers annealing in the respective upstream exon and unlabeled primers annealing in the respective downstream exon

(Supplementary Table S1). Three examples, i.e., splicing of pre-mRNAs for FAM90A1, CDC6, and LETM2, are shown in Fig. 6d (complete gels in Supplementary Fig. S6). Quantification (Fig. 6e) confirmed a significant increase in intron retention in these pre-mRNAs upon rCD1 treatment compared to the DMSO-treated controls. These observations suggest that partial recruitment of PRPF38A to the nuclear lamina results in moderately increased intron retention, reflective of a moderate downregulation of splicing efficiency.

## Discussion

To target spliceosomal proteins to distinct subcellular or subnuclear regions, we examined the suitability of proteins with different cellular localizations to function as anchors for rCID-dependent recruitment, using PRPF38A and U2AF35 as exemplary target splicing factors. To this end, we fused an FKBP domain (alone or in combination with RFP) to the N-termini of the targets and used anchor-ECFP-SNAP fusions as anchor constructs.

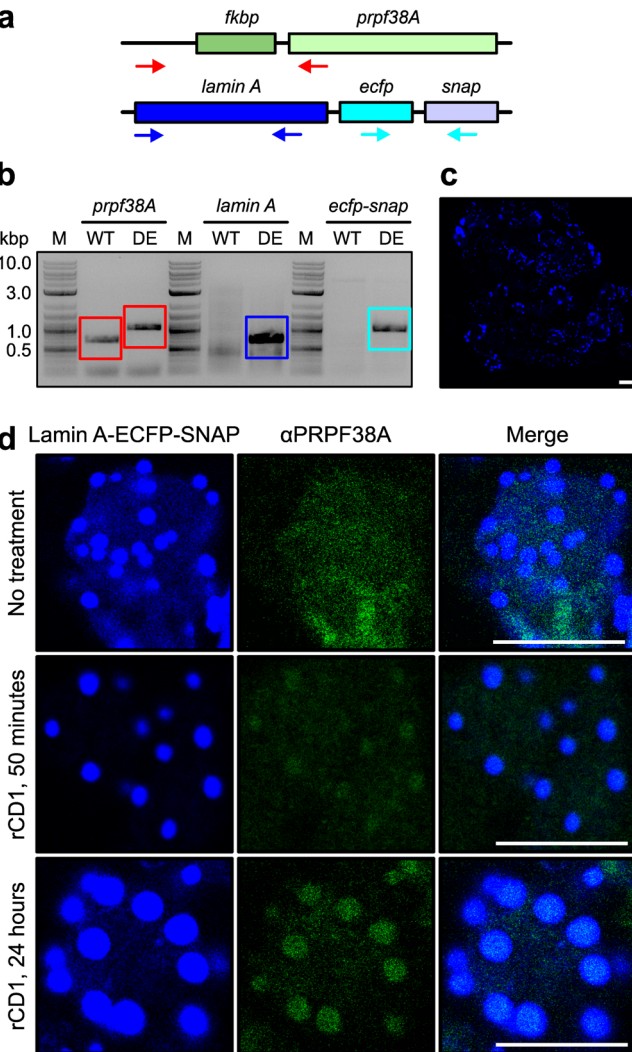

**Fig. 5 Generation of a doubly engineered FKBP-PRPF38A CRISPR/lamin A-ECFP-SNAP stable cell line. a** Schemes depicting primer pairs (arrows; Supplementary Table S1) used in the experiments shown in **b**. **b** PCR analysis of the genomic DNA of a doubly engineered FKBP-PRPF38A CRISPR/lamin A-ECFP-SNAP stable cell line, revealing the presence of the insertion of the FKBP-coding region and additional stable integration of the lamin A-ECFP-SNAP-coding region. Amplified products are boxed in colors corresponding to the primer pairs in **a**. WT wild-type HEK293T cells; DE doubly engineered FKBP-PRPF38A CRISPR/lamin A-ECFP-SNAP stable cells; M, size marker (kbp). **c** Immunofluorescence analysis of doubly engineered FKBP-PRPF38A CRISPR/lamin A-ECFP-SNAP stable cells, showing an ECFP fluorescence signal (blue channel) consistent with the expression of the lamin A-ECFP-SNAP fusion protein. A scale bar (10 μm) is shown as a white line on the bottom right of the image. **d** Untreated (top row) or rCD1-treated (middle row, 50 min; bottom row, 24 h) doubly engineered FKBP-PRPF38A CRISPR/lamin A-ECFP-SNAP stable cells were stained with anti-PRPF38A primary antibody and an Alexa488-coupled secondary antibody. Z-stacks were recorded, revealing the co-localization of FKBP-PRPF38A with lamin A-ECFP-SNAP at spherical lamin A-ECFP-SNAP aggregates only after rCD1 treatment. Scale bars (10 μm) are shown as white lines on the bottom right of the merged images.

Whereas rCD1-dependent recruitment to the PM or the endo-membrane system (anchor proteins Lck-ECFP-SNAP or ECFP-SNAP-CAAX, respectively) was not successful, rCD1 induced the recruitment to nucleoli (anchor protein RPL13-ECFP-SNAP) or the nuclear lamina (anchor protein lamin A-ECFP-SNAP). As even

a long-time incubation with rCD1 (up to 24 h) did not result in the recruitment of splicing factors to the PM, one requirement for the successful re-localization of splicing factors, thus, seems to be that the anchor protein resides in the same cellular compartment for extended times. It is possible that rCD1-induced dimerization of target proteins and anchor proteins localized at the PM happened during cell division upon nuclear membrane disassembly and that it was disrupted again after nuclear membrane reformation. In this case, the nuclear localization signal of the spliceosomal protein might have overcome rCID-dependent PM-localization.

rCID was also differentially effective in targeting splicing factors in different sub-nuclear regions. Splicing factors continuously cycle between Cajal bodies, speckles, the nucleoplasm, and peri-chromatin[9,22]. Cajal bodies and speckles themselves represent highly dynamic structures, undergoing movements within the inter-chromatin space and associating with actively transcribed genes and chromatin regions[21,22,25,60,61]. In contrast, nucleoli are involved in very few splicing-related metabolic processes, pre-dominantly the modification of U6 snRNA, and spliceosomal components rarely locate to nucleoli[62], possibly lowering the efficiency of the recruitment to nucleoli via rCID. In contrast, spliceosomal proteins can diffuse to chromatin regions either directly through the nucleoplasm or from Cajal bodies and speckles, which are in contact with chromatin. The lamina is also in direct proximity to chromatin[63], especially with lamina-associated domains of chromatin that are correlated with low gene expression levels[42,43]. This topology may facilitate the diffusion of splicing factors from nucleoplasmic regions to the lamina, improving the efficiency of rCID-based recruitment.

The relative expression levels of target and anchor protein fusions was observed as another important determinant of rCID efficiency. Recruitment only worked efficiently with an excess of the anchor protein fusion, probably indicative of a mass action effect. A low, wild-type-like expression level of the targeted spliceosomal protein is most likely beneficial to elicit splicing effects by rCID, as it will reduce the level of the non-recruited pool that is still available for splicing. Insertion of an FKBP-coding region at the endogenous genetic locus of the spliceosomal target protein by CRISPR/Cas9-based genome engineering will sustain the endogenous expression level of the targeted splicing factor. Especially for the analysis of splicing factors, an endogenous expression level is also important to avoid the re-distribution of the factors among the sub-nuclear regions, such as Cajal bodies, speckles, inter-chromatin space, and peri-chromatin fibrils, due to an altered protein level. E.g., overexpression of splicing factors can shift the equilibrium to increased storage in nuclear speckles[23,24], and the kinetics of splicing events them-selves were shown to be sensitive to splicing factor levels[64]. In summary, we conclude that efficient rCID-based re-localization of splicing factors requires the localization of the target and anchor protein constructs to the same cellular membrane-bound compartment (or possibly a rCID system that can compete with the differential subcellular targeting of target and anchor protein constructs) and an excess of the anchor over the target protein construct.

Recruitment of FKBP-PRPF38A to lamin A-ECFP-SNAP at the lamina, with a residual pool of untagged PRPF38A being unaffected, induced a moderate increase in intron retention and only very minor or no effects on other alternative splicing events. Clearly, the effects observed in our setup are attenuated by the pool of non-recruitable PRPF38A that remains probably due to proteolytic cleavage of the fused FKBP tag within the cells. This problem may be alleviated by fusing an FKBP tag to the PRPF38A C-terminus, which may be less prone to protease cleavage. Alternatively, the fusion of FKBP tags to both PRPF38A termini might ensure that all endogenous PRPF38A molecules retain at

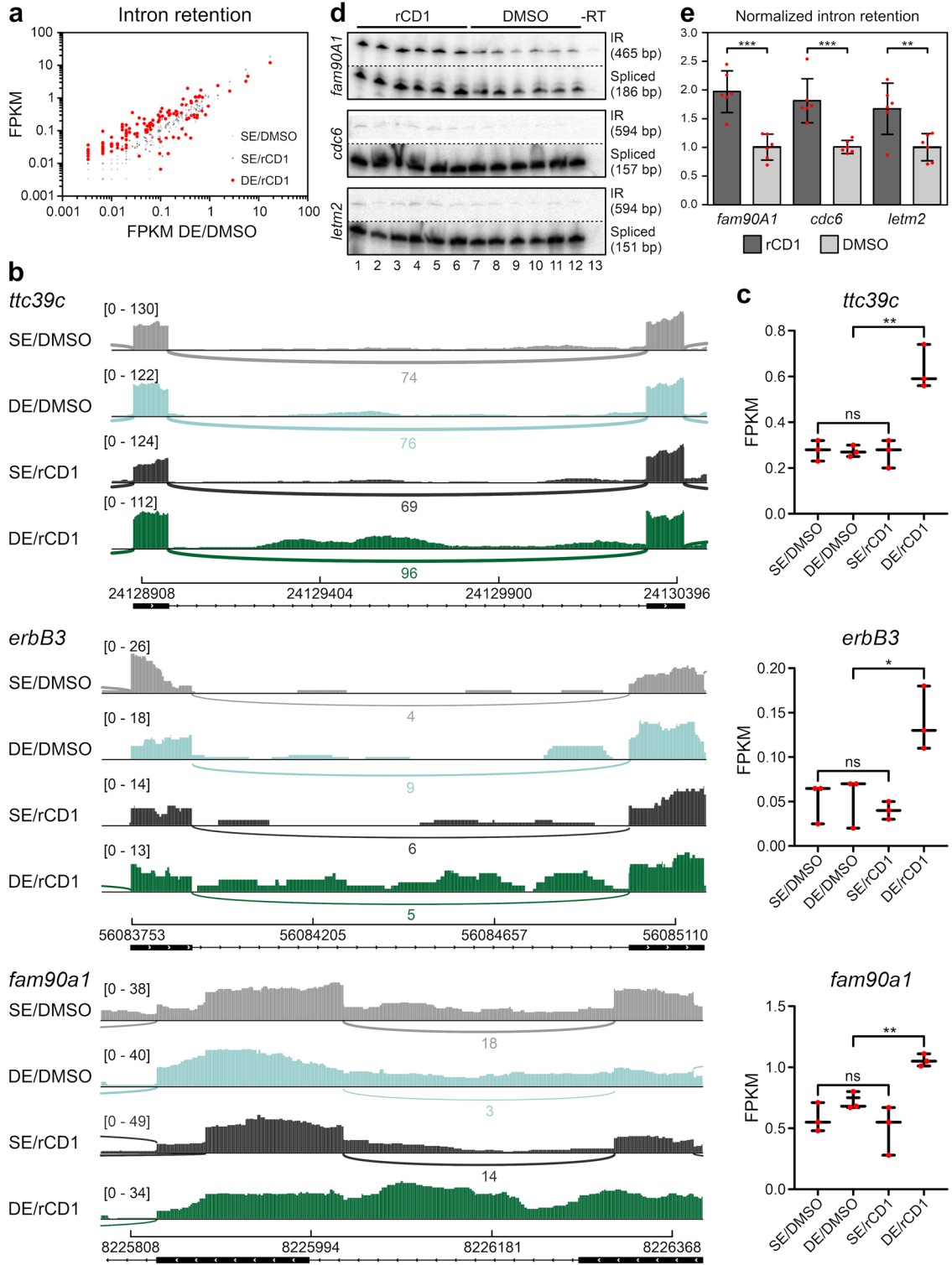

least one tag via which they can be reversibly anchored away. The trend of increased intron retention is still clearly significant and indicative of a reduced splicing capacity of the cells after rCD1 treatment. As PRPF38A is a known interaction hub with a multitude of protein–protein interaction partners[33–36], its recruitment to the lamina might additionally carry other interacting spliceosomal factors along, which could have exacerbated the increase in intron retention observed in our assays. Our observations are in agreement with a previous study, in which the knockdown of PRPF38A caused a significant increase in intron

retention, but did not drastically alter exon skipping, mutually exclusive exons, or alternative splice site usage[65]. This observation suggests that rCID-dependent splicing factor recruitment to the nuclear lamina offers an alternative to siRNA-mediated knockdown that can quickly reduce active levels of the targeted factor and that can be reversed.

Human PRPF38A joins the spliceosome as part of the B complex[31,32] and it comprises a C-terminal RS-like domain, which is only present in higher eukaryotes and absent in *Saccharomyces cerevisiae*[36]. RS-domains are a hallmark of the splicing regulatory

**Fig. 6 RNAseq analysis and validation.** Singly engineered FKBP-PRPF38A CRISPR cells (SE) or doubly engineered FKBP-PRPF38A CRISPR/lamin A-ECFP-SNAP stable cells (DE) were treated with DMSO (SE/DMSO or DE/DMSO) or rCD1 (SE/rCD1 or DE/rCD1). Cellular RNAs were sequenced ($n = 3$ biological replicates) and sequencing results were evaluated with the iREAD tool. **a** Fragments per kilobase of transcript per million mapped reads (FPKM) values of introns, whose retention was significantly altered ($p < 0.001$) in the DE/rCD1 samples (red) plotted against FPKM values of DE/DMSO samples. The FPKM values for the controls SE/DMSO (light gray) and SE/rCD1 (dark gray) are plotted against FPKM values of DE/DMSO for comparison. **b** Sashimi plots showing three examples of enhanced intron retention (*ttc39c, erbB3, fam90A1* pre-mRNAs) in doubly engineered FKBP-PRPF38A CRISPR/lamin A-ECFP-SNAP stable cells after rCD1 treatment (dark green), compared to the controls (SE/DMSO, light gray; DE/DMSO, light green; SE/rCD1, dark gray). **c** Box plots reporting FPKM values from the RNAseq data of the genes illustrated in the sashimi plots in **b**. Significance according to a *t*-test between the conditions is indicated with ns for not significant, * for $p \le 0.05$, ** for $p \le 0.01$, and *** for $p \le 0.001$. The *t*-test showed significance between the treatment groups for *ttc39c* ($p = 0.003$), *erbB3* ($p = 0.03$) and *fam90A1* ($p = 0.003$). **d** PCR analyses of $n = 6$ biological replicates were performed with radioactively labeled forward primers annealing to the respective upstream exon and unlabeled primers annealing to the respective downstream exons of the *fam90A1, cdc6,* and *letm2* genes in doubly engineered FKBP-PRPF38A CRISPR/lamin A-ECFP-SNAP stable cells, showing enhanced intron retention after rCD1 treatment (lanes 1–6) compared to DMSO-treated cells (lanes 7–12). -RT, PCR analysis without the addition of reverse transcriptase upon cDNA generation (control for the background generated from undigested genomic DNA). The nature of the PCR products and their sizes are indicated on the right. IR, intron-retained product. Relevant regions of the same gels were excised for display purposes as indicated by the dashed lines. Full gels are shown in Supplementary Fig. S6. **e** Quantification of the data shown in **d**. Data indicate means of normalized intron retention levels ± SD of six biological replicates. Intron retention levels were calculated as the ratio of the intensity of the band representing the intron-containing pre-mRNA and the sum of the intensities of the bands representing the intron-containing pre-mRNA and spliced product. Normalized intron retention levels were calculated as the ratio of intron retention levels and the mean intron retention level upon DMSO treatment. Individual replicate values are shown in red. Significance according to a *t*-test between the DMSO and rCD1-treated replicates is indicated with ** for $p \le 0.01$ and *** for $p \le 0.001$. The *t*-test showed significance between the treatment groups for *fam90A1* ($p = 0.0002$), *cdc6* ($p = 0.0005$), and *letm2* ($p = 0.008$).

SR proteins[66]. Several B-specific proteins[67–70] and SR proteins[71] have been implicated in alternative splicing. However, our findings in conjunction with the observations by Chan et al.[65] rather support the notion that PRPF38A can be considered a constitutive splicing factor that has only a minor role in regulating alternative splicing. Based on recent cryoEM structures of the U4/U6-U5 tri-snRNP as well as the B and the B$^{act}$ spliceosomal complexes, it was hypothesized that PRPF38A triggers conformational changes of the spliceosome required for activation. When the PRPF8 core and N-domain change from an open conformation in the tri-snRNP to a partially closed conformation in the B complex, human PRPF38A probably locks the RNA-containing exon channel between these domains to stabilize the closed conformation of PRPF8[72]. This function of PRPF38A is likely required for every splicing event.

Mis-splicing is a frequent principle of genetic diseases and an attribute of many cancers[73,74]. Therefore, there is a high demand for versatile methods to study the complex mechanisms of splicing and splicing regulation. Traditional methods for gene-function analyses and gene therapies are based on antisense technologies. Splicing factors or mutant transcripts can be targeted by siRNAs or shRNAs to alter levels of specific mRNAs[75]. However, RNAi-based knockdown approaches can elicit secondary or off-target effects due to imperfect complementary of the small RNAs to other cellular RNAs[76–78]. Antisense morpholino oligonucleotides offer an alternative to RNAi techniques, but may also suffer from off-target effects due to sequence complementarity to other RNAs or by the activation of cryptic splice sites[79]. rCID-based splicing factor manipulation may have fewer side effects, as suggested by the negligible consequences on pre-mRNA splicing by rCD1 treatment of the control cell line without recruitment we observed. The dimerizer rCD1 specifically binds to the introduced tags and likely does not bind tightly to many other cellular targets, except endogenous FKBP. Potential off-target or side effects can be assessed with simple controls. Potential side effects of transient over-expression of target or anchor protein constructs can be circumvented by inserting the target protein tag into the genome via CRISPR/Cas9-based genome engineering and the anchor protein construct via stable integration. In case of a highly expressed anchor protein, additional CRISPR/Cas9-based fusion to a SNAP-tag may also be an option. This strategy also alleviates potential side effects of transfections, such as interferon responses and autophagy[80,81]. While small molecule inhibitors targeting specific splicing factors would in principle also allow for their swift manipulation, the development of such inhibitors is time-consuming and some proteins may not exhibit suitable binding pockets or interaction surfaces for such molecules.

Apart from the rCD1-based method presented here, other chemical dimerizer-based techniques have been reported for targeted protein re-localization. One of the first chemical dimerizer-based "anchor-away" systems developed was based on rapamycin, which mediates the dimerization of FKBP12-tagged and FRB-tagged proteins[82], and which was used to recruit nuclear proteins to distinct cellular locations in yeast, e.g., the cytoplasm via tagged RPL13A[83]. The same approach was subsequently applied in a proof-of-principle study in *Drosophila melanogaster* to study loss-of-function phenotypes of nuclear proteins[84]. A similar strategy has been applied in mammalian cells, involving abscisic acid-mediated dimerization of phytohormone factor fusion proteins, with the results suggesting that it could be used to probe the function of chromatin-modifying proteins with reduced side effects[85]. However, most of these alternative strategies either require extensive washout to reverse the effects or the effects cannot be efficiently reversed due to a high affinity of the chemical dimerizer to the target protein fusion (e.g., in the case of rapamycin)[86]. As rCD1-based recruitment can be rapidly reversed, the approach presented here adds additional versatility to the chemical dimerizer-based toolbox for nuclear proteins.

In the abovementioned studies, yeast RPL13A or the *Drosophila*/ human homologs RPL13 were used as anchors to mis-localize nuclear proteins to the cytoplasm, as ribosomal proteins are imported into the nucleus after their production in the cytoplasm, where they are assembled into ribosomal complexes that are subsequently exported again to the cytoplasm. In our study, over-production of RPL13-ECFP-SNAP led to an accumulation of the anchor fusion construct in nucleoli, presumably because it was not efficiently or completely integrated into ribosomal complexes for nuclear export. While CRISPR/Cas9-based SNAP-tagging of endogenous RPL13 may in principle prevent these effects, homozygous tagging of RPL13 could not be achieved in a past study[85], possibly because of loss of RPL13 functionality in the ribosome.

In addition to RNAi-based strategies, targeted protein degradation offers additional possibilities for loss-of-function analyses. Although the degradation of a target protein can be achieved

within hours for many degradation systems (e.g., about 1–8 h for the dTAG system[87] and about 4–8 h for the HaloPROTAC system[88]), fast degradation of some proteins within 20–30 min has been described (e.g., using auxin-inducible degradation technologies[89,90]). The degradation kinetics depend on the target protein subjected to the degradation and can usually not be predicted. Irrespectively, the reversion of the depletion, if possible, is even much slower, in contrast to the method presented here.

Key benefits of rCID-based splicing factor manipulation are the fast re-localization kinetics and a high level of temporal control. Cells rapidly respond to changing environments by activating signaling cascades, and several studies indicated that alternative splicing is frequently used to induce specific cellular responses[91]. In contrast to transcriptional regulation, regulation on the post-transcriptional and post-translational levels is fast. Indeed, some recent studies implicated alternative splicing in the regulation of immediate-early responses induced shortly after cellular stimulation, e.g., in the activation of T-cells and after neuronal stimulation[92–94]. One possible mechanism for such immediate-early responses is posttranscriptional splicing of intron-containing transcripts. Although intron retention was previously considered to be rare in mammals, surprisingly, 50% of all introns were recently reported to show detectable intron retention in many human and mouse cell types[95]. Furthermore, it was proposed that some pre-mRNAs are retained in the nucleus for posttranscriptional splicing[24] and that this posttranscriptional splicing might even occur in speckles[96]. To study such fast events, swift manipulation methods are required. Effects elicited by siRNAs, shRNAs, or morpholinos can usually be discerned only after days in cell culture experiments. Furthermore, the effects are not readily reversible. Reversion of the effects of small molecule inhibitors can also be difficult and would typically involve extensive washout. As shown here, the recruitment of FKBP-PRPF38A to the lamina was achieved within around 30–40 min after rCD1 treatment, rendering the system suitable to analyze fast splicing switches. As the dimerization by rCD1 can be quickly reversed upon incubation with FK506, transient splicing changes can be more accurately mimicked than with other methods.

Taken together, the nuclear topology forms the basis for another level of splicing regulation, which is still largely unexplored. Further development of rCID-based splicing factor re-localization, e.g., by employing yet other anchors, may offer an inroad into this level of regulation by allowing the controlled recruitment of splicing factors to diverse sub-nuclear regions with high spatial precision and in a temporally controlled manner.

## Methods

### Genetic constructs, cell culture, transfection, and chemical dimerizer treatment.

DNA fragments encoding anchor proteins large ribosomal subunit protein eL13 (RPL13), heterochromatin protein 1 homolog alpha (HP1) and lamin A were PCR-amplified from the MegaMan Human Transcriptome Library (Agilent Technologies). DNA fragments encoding target splicing factors PRPF38A and U2AF35 were PCR-amplified from genetic constructs used in previous studies[34,97]. Vectors guiding expression of the constructs Lck-ECFP-SNAP, ECFP-SNAP-CAAX, and RFP-FKBP (Lck, leukocyte C-terminal Src kinase; ECFP, enhanced cyan fluorescent protein; RFP, red fluorescent protein; FKBP, FK506-binding protein 12) were described previously[28,37,98]. The vectors for expression of other anchor and target protein fusions were obtained by Exponential Megapriming PCR (EMP) cloning[99] into the above vectors.

HEK293T cells (Leibniz-Institut DSMZ) were cultivated at 37 °C and 5% CO₂ in DMEM Glutamax high glucose medium (Gibco), supplemented with 10% fetal bovine serum (Biowest). For transfection, cells were seeded and treated with Lipofectamine 2000 (Invitrogen) according to the manufacturer's instructions. Plasmids expressing anchor-ECFP-SNAP fusion proteins were transfected in three-fold molar excess to plasmids expressing target-RFP-FKBP fusion proteins for subsequent dimerization of over-expressed proteins[37]. Knockdown of endogenous PRPF38A was performed with the siRNA 5′-GGAUAUCAUUGUAGAGUUUdTdT-3′ (eurofins genomics); the siRNA 5′-UUUGUAAUCGUCGAUACCCdTdT-3′ was used as a control. The siRNAs were transfected with 50 nM final concentrations. To

induce rCID, cells were treated with 6 µM rCD1[28]. To reverse the dimerization, 6 µM FK506 (Tocris Bioscience) were added to the cells.

### Western blot.

Cells were washed with phosphate-buffered saline (PBS), centrifuged and re-suspended in radioimmunoprecipitation assay (RIPA) buffer (20 mM Tris-HCl, pH 8.0, 2% [v/v] NP-40, 10 mg/ml sodium desoxycholate, 4 mM EDTA, 200 mM NaCl), supplemented with protease inhibitors (cOmplete tablets, Roche). The protein concentration was determined via a Bradford assay and equal amounts were separated by SDS-PAGE. Proteins were transferred to a PVDF membrane via a semidry blotting method. Detection was done with the following antibodies: anti-PRPF38A NBP2-37697 antibody (Novus Biologicals) 1:2000 diluted or NBP2-33673 antibody (Novus Biologicals) diluted to 0.4 µg/ml; anti-actin MAB1501 antibody (Merck Millipore) 1:4000 diluted; anti-GFP sc-9996 antibody (Santa Cruz) 1:1000 diluted; goat anti-rabbit 31460 antibody (Thermo Fisher Scientific) 1:10000 diluted; goat anti-mouse 31430 antibody (Thermo Fisher Scientific) 1:5000 diluted. Membrane stripping was done by incubation of the membrane in stripping buffer (Glycin 0.2 M, SDS 1 % (w/v), Tween-20 0.1% (w/v), pH 3.2) at 50 °C for 1 h.

### RNA extraction, reverse transcription, and radioactive PCR.

After washing the cells with PBS, cells were re-suspended in Trizol (RNATri, Bio&SELL) and the total RNA was extracted by phenol-chloroform extraction, isopropanol precipitation, DNase digestion, and ethanol-precipitation. The RNA was reverse transcribed with gene-specific reverse primers by MuLV reverse transcriptase (Qiagen). Radioactive PCRs were performed with [³²P]-labeled forward primers and unlabeled reverse primers (Supplementary Table S1) and the products were separated on a denaturing polyacrylamide gel (4 or 5% acrylamide, 7 M urea, 0.5 x TBE). The gel was transferred to a filter paper, dried, and radioactive bands were detected on a Typhoon phosphorimager (GE Healthcare) via a storage phosphor screen after exposure to the filter paper. Quantification was performed using ImageQuant (GE Healthcare). The intron retention level was calculated by the ratio of the intensity of the intron-containing band to the sum of the intensities of the spliced and intron-containing bands. Normalization was done by dividing the intron retention values of each sample by the average of the intron retention of all DMSO-treated samples of the respective target. Selected PCR amplification products were cloned into the pJET 1.2 vectors with the pJET 1.2. cloning kit (Thermo Fisher Scientific) and sequenced via Sanger sequencing (Microsynth Seqlab).

### Confocal microscopy.

For live-cell imaging, cells were grown and imaged in 8-well chamber slides (80826, Ibidi). For imaging of fixed cells, cells were grown on sterilized coverslips, washed with PBS, and fixed in a 3.7% (v/v) formaldehyde solution for 20 min. The cells were permeabilized with 0.1% (v/v) Triton X-100 for 20 min, blocked with 5% (w/v) non-fat dry milk in Tris-buffered saline with Tween-20 (TBS-T), and incubated with primary anti-PRPF38A NBP2-37697 antibody, diluted 1:50 in 2.5% (w/v) non-fat dry milk in TBS-T, for 2 h. After washing, the slips were incubated for 1 h in secondary goat anti-rabbit Alexa Fluor 488 A-11008 antibody (Thermo Fisher Scientific), diluted 1:500 in 2.5% (w/v) non-fat dry milk in TBS-T. The cells were mounted on cover slides with one drop of self-prepared Dabco Mowiol. Living and fixed cells were imaged on a Leica TC SP8 microscope with a 63 x magnification immersion oil objective, controlled by the LAS X software (Leica). Images were exported from the LAS X software as TIFF files and imported into CorelDRAW 2019 for figure generation; in case the blue fluorescent signal was too weak to be clearly observed in the emerging figure, an enhanced contrast was applied to the whole image with the LAS X software. For quantification, raw images were exported as TIFF files and imported into the Fiji image processing package[100]. The RFP and ECFP channels were split and images were converted to gray-scale. Following background subtraction, regions of interest (ROIs; nuclei of cells expressing both target and anchor protein) were manually defined. The degree of co-localization was quantified via the co-localization threshold plugin of Fiji. The $R_{coloc}$ values of the output were used to estimate the degree of recruitment. $R_{coloc}$ values for individual ROIs were normalized by subtraction of the $R_{coloc}$ value of the respective ROI for the first time point after rCD1 addition (two minutes) and division by the corresponding $R_{coloc}$ value for the respective ROI 40 min after rCD1 addition.

### Genomic engineering.

For CRISPR/Cas9-based fusion of a DNA fragment encoding the FKBP tag to the *prpf38a* locus, suitable guide RNAs (gRNAs) were identified in silico (Supplementary Table S2) and cloned into the PX459 2.0 vector (plasmid 62988; Addgene). For a cleavage efficiency test, the respective gRNA vectors were transfected with control gRNA vectors in a 1:1 ratio, incubated for 3 days, and selected with 3 µg/ml puromycin for 3 days. The genomic DNA of the cells was extracted and proteins were digested with 0.1% (v/v) gelatin, 8% (v/v) Q5 PCR reaction buffer, 4% (v/v) Tween-20, 0.4% (v/v) NP-40, 0.04 mg/ml proteinase K. The DNA was used for PCR amplification with primers also used for testing insertion of the FKBP-coding region on the *prpf38A* locus (Supplementary Table S1). For the knock-ins, a repair template with the FKBP-coding sequence 5′ of the PRPF38A start codon was designed with a 1500 nucleotide (nt) 5′-overhang, 1000 nt 3′-overhang, and a mutated PAM site (ordered from Thermo Fisher Scientific), linearized and co-transfected with gRNA3 vector into the cells. The

selection was done as described above. Colonies were obtained from single cells by dilution and tested for integration by PCR amplification of extracted genomic DNA with suitable primers. The CRISPR cell line C3 with a mutated start codon (methionine to leucine) was obtained by editing the CRISPR cell line C1 with gRNA4 and a mutated repair template.

For the generation of a stable cell line, the DNA fragment encoding a lamin A-ECFP-SNAP fusion protein was cloned into the pcDNA 3.1 vector (plasmid 87063; Addgene) containing a hygromycin resistance gene. The HEK293T CRISPR C1 cells were transfected with the linearized lamin A-ECFP-SNAP-encoding pcDNA 3.1 vector, incubated for 2 days, and selected with 300 μg/ml hygromycin B for 2 weeks. Colonies were obtained from single cells by dilution and were inspected by PCR of the genomic DNA with suitable primers, as described above. The stable cell line was further cultivated in 200 μg/ml hygromycin B and seeded for assays without hygromycin B.

**RNA sequencing**. The total RNA was extracted from FKBP-PRPF38A CRISPR C1 cells (referred to as "singly engineered" [SE] cell line below) or from FKBP-PRPF38A CRISPR C1 cells that additionally stably expressed lamin A-ECFP-SNAP fusion protein (referred to as "doubly engineered" [DE] cell line below) after treatment with rCD1 or DMSO (rCD1 solvent) for 24 h. RNA sequencing of poly(A)-selected RNAs was performed in biological triplicates yielding ~50 million paired-end 150-nt reads per replicate. Reads were aligned to the human hg38 genome using STAR, resulting in ~75% of uniquely aligned reads. For the following iREAD analysis, the aligned bam files were indexed using bamtools. Additionally, triplicate bam files were merged and indexed with bamtools. The resulting bam files (per condition) were visualized using the IGV browser. In sashimi plots, minor splicing isoforms were excluded. Intron retention levels were calculated using iREAD. In iREAD, introns were considered differentially spliced with a $t$-test-derived $p$ value <0.001 between the FPKM intron values of triplicate rCD1-treated DE samples and the nine control samples. Furthermore, alternative splicing changes were calculated using RMATS. To focus on high-confidence targets, only targets with a $p$ value <0.001 and a deltaPSI >0.1 were considered alternatively spliced. Additionally, to filter out splicing events in poorly expressed genes or gene regions with low expression, we excluded events with less than 100 combined junction reads in all samples. For sashimi plots (Fig. 6b), "junction read coverage min" was set to 3–5 per track. For quantifications, iREAD-derived FPKM values were plotted in GraphPad Prism as box-whisker plots including individual data points (horizontal lines, medians; whiskers, minimum, and maximum values).

**Statistics and reproducibility**. Experiments subjected to significance analysis were performed at least in triplicates ($n \geq 3$). Significance was assessed via unpaired, two-sided, homoscedastic $t$-tests; significance indicators: ns not significant; *$p \leq 0.05$; **$p \leq 0.01$; ***$p \leq 0.001$. Specific $p$ values are reported in the respective figure legends.

**Reporting summary**. Further information on research design is available in the Nature Research Reporting Summary linked to this article.

## Data availability

RNAseq data were deposited at Gene Expression Omnibus (https://www.ncbi.nlm.nih.gov/geo) under accession code GSE182412[59]. All other data supporting the findings of this study are described in the manuscript or in the Supplementary Data or are available from the corresponding authors on request. Biological materials (plasmids, vectors, and cell lines) are available from the corresponding authors upon request.

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

## Acknowledgements

We thank Katharina Achazi, Freie Universität Berlin, for support in the organization of the cell culture and microscopy facilities. We would like to acknowledge the assistance of the BioSupraMol core facility of Freie Universität Berlin, supported by the Deutsche Forschungsgemeinschaft. This work was funded by grants from the Deutsche Forschungsgemeinschaft (TRR186/Mercator Fellow to C.S. and TRR186/A15 to F.H. and M.C.W.).

## Author contributions

K.V. performed experiments. M.P. performed bioinformatics analyses of RNAseq results. N.H. helped in molecular cloning. S.F. and C.S. prepared rCD1. All authors participated in data interpretation. K.V. and M.C.W. wrote the manuscript with input from the other authors. C.S., F.H. and M.C.W. supervised work in their respective groups. F.H. and M.C.W. conceived and coordinated the project. C.S., F.H. and M.C.W. provided funding for the project.

## Funding

## Competing interests

The authors declare no competing interests.
