## [Peer Review File · Communications Biology]

Reviewers' comments:

Reviewer #1 (Remarks to the Author):

This manuscript "Recruitment of a splicing factor to the nuclear lamina" describes the development of a technique to tether a nuclear protein to a biologically inactive region, thereby disrupting the function of the protein. I would consider it a variant of the "anchor-away" method that was developed in yeast some time ago. Specifically, Markus Wahl and co-authors describe the inducible relocalisation of PRPF38A to the nuclear lamina. For this purpose, they tag PRPF38A with an FKBP tag and express a SNAP-tagged lamin A. The interaction between the two proteins is induced by a chemical dimerizer rCD1. As a result, PRPF38 is bound to the lamina within a short period of time and is no longer available for the activation of the spliceosome. This results in a detectable drop in splicing activity as measured by increased intron retention. The work is, as the authors themselves write, more a proof-of-principle study than the comprehensive development of a new method and its application to different biological questions. As described in the manuscript, the authors faced several problems and had to overcome some obstacles. For example, it proved impossible to tag and relocalise the entire pool of PRPF38A protein. Some of the PRPF38A lost its tag and thus could not be "controlled". This is probably also the reason why the functional effects were not very pronounced.

The manuscript is written very well and is a pleasure to read. The experiments are well controlled and the results, with a few exceptions (see below), are presented appealingly. Certainly, the method is interesting mainly for a limited audience, but it represents a good alternative for the functional inactivation of nuclear proteins that cannot be depleted by other methods.

Specific comments:

1. This is probably too much to ask for in such a proof-of-principle study - but it would of course be great if the authors could show the effect of relocalization with another protein. U2AF35 would be an obvious choice here, as it has been shown to be well-behaved in the authors' preliminary experiments. Perhaps the authors have already worked on an endogenously tagged U2AF35 and first results are already available?
2. The title is not quite clear. I think it should confer the message that the process of recruitment to the nuclear lamina leads to the inactivation of the splicing factor.
3. I am not too happy with the way the data are shown in Figure 6.
 - a. The illustration in Figure 6b can certainly be improved and possibly also include some sort of quantification of the RNA-Seq data.
 - b. Are the validation PCRs in Figure 6c really the best that the authors can show? For example, I find it hard to believe that the band for cdc6 in lane 3 can be accurately quantified.
 - c. I find it surprising that Figure 6d was normalized to the rCD1 condition. I would have rather set the DMSO control as 1 to show the intron retention as an increase.

Reviewer #2 (Remarks to the Author):

Vester et al. elucidate how to mislocalize a splicing factor from its native localization to perturb its function in order to study its effect on mRNA splicing. To this end, they employ reversible chemically induced dimerization. Upon induction, target protein and different anchor proteins are dimerized, which should lead to a change of localization of the target protein to match the localization of the anchor protein. The authors evaluate several possible anchor proteins and show that the best effect is achieved with lamin A, leading to localization of target protein to the nuclear lamina within less than an hour with possible reversion in a few minutes.

This work introduces a useful technology into the field of splicing, which will likely enable many new insights. This is highlighted by novel findings about the effect of mislocalization of splicing factor PRPF38A, which results in changes in intron retention, mostly enhancing intron retention.

The study was conducted in a scientifically sound manner and is well presented in a logical and easy-to-follow fashion. The manuscript is well written.

I strongly appreciate that different difficulties in establishing the system are discussed within the manuscript, which is very beneficial for potential users of this technique.

There are a few questions and minor issues that should be addressed in my opinion. They are presented here in order of appearance in the text:

- 1) Please mention the study organism/model in the abstract, to make it easier to grasp.
- 2) l. 161 and following: the fact that the mislocalization of PRPF38A also scales with transfection efficiency of the anchor construct (as seen by fluorescence remaining in the nucleoplasm) does not become clear to me within the text, but is mentioned in the figure caption. It would be good to state this more clearly in the text as well.
- 3) l. 254 and following: The remaining pool of PRPF38A without FKBP is a problem that may prevent seeing more drastic effects. It would be good if the authors could point to how this issue might be improved. E.g., would it be possible to add a second, C-terminal tag, or to remove the unstructured N-terminus to enable more complete re-localization? In my opinion this does not call for additional experiments but should be discussed.
- 4) l. 267: The authors state: "To obtain a higher expression of the anchor protein fusion over the target protein fusion, we stably integrated the lamin A-ECFP269 SNAP-coding region into the genome of the C1 CRISPR cell line."
The stable integration itself only warrants higher expression of anchor protein fusion over target protein if the construct is indeed expressed more strongly than the target. Was this tested, and if so, how? Or was a specific promoter chosen to ensure high expression? This could be stated more clearly.
- 5) In the discussion, I felt that some points might be missing. Specifically, since the method resembles the Anchor-away strategy (Haruki et al., 2008) that has been used in yeast to achieve re-localization of nuclear factors from the nucleus to other compartments after induced dimerization, I would have expected a mention of this technique. Also, the strategy has recently been adapted to mammalian and other non-yeast cells in a few instances (Galloy et al., 2021, Sanchez Bosch et al., 2020, I may have missed others). I feel that those other approaches should be acknowledged and compared to the method developed here. Might these methods even be orthogonal?
- 6) In particular, Galloy et al. 2021 used an RPL13 fusion to successfully mislocalize a nuclear protein. I would be curious about possible reasons why RPL13 fusion was less successful in the approach presented here compared to Galloy et al.
- 7) L. 445: Targeted depletion is described as very slow, taking "at least 4-8 hours" to achieve degradation for most cases. However, descriptions of the auxin induced degron system in mammalian cells state rapid depletion with half-lives of 20-40 minutes for the standard AID and around 20 min for AID2 (Yesbolatova et al., 2020). In the latter, it looks like the vast majority of protein will be degraded after 90 minutes. Also, the cited paper by Habet et al. on the dTAG system shows robust degradation to baseline levels of many of the proteins tested within 1 hour (Habet et al., Fig. 4a). Therefore, I believe the authors should reconsider the representation of these methods. A main advantage of the new system presented here clearly is the possibility of very fast recovery of protein localization within few minutes. To me it is not clear whether the 2 papers cited here at the end of the sentence at l. 446 refer to degradation time or recovery time (I agree recovery is much slower in degradation methods). If the authors are aware of particularly slow degradation methods that are widely used, they might want to include citations directly after "at least 4-8 hours" in l. 445.
- 8) In the Methods section dilutions of secondary antibodies for the Western Blots are not stated at l. 491-492.
- 9) L. 503: I assume the bands were not detected directly but a storage phosphor screen was exposed to the dried filter paper and then scanned on the Typhoon scanner.
- 10) L. 576: What are treat_stbl samples?

11) L. 585: For the T-tests it is not stated what type they were (1- or 2-tailed, homoscedastic or heteroscedastic).

Response to Reviewer Comments

Reviewer comments are repeated in bold italics, responses are in regular font, cited text passages are highlighted in yellow.

Reviewer #1 (Remarks to the Author):

This manuscript “Recruitment of a splicing factor to the nuclear lamina” describes the development of a technique to tether a nuclear protein to a biologically inactive region, thereby disrupting the function of the protein. I would consider it a variant of the “anchor-away” method that was developed in yeast some time ago. Specifically, Markus Wahl and co-authors describe the inducible relocalisation of PRPF38A to the nuclear lamina. For this purpose, they tag PRPF38A with an FKBP tag and express a SNAP-tagged lamin A. The interaction between the two proteins is induced by a chemical dimerizer rCD1. As a result, PRPF38 is bound to the lamina within a short period of time and is no longer available for the activation of the spliceosome. This results in a detectable drop in splicing activity as measured by increased intron retention. The work is, as the authors themselves write, more a proof-of-principle study than the comprehensive development of a new method and its application to different biological questions. As described in the manuscript, the authors faced several problems and had to overcome some obstacles. For example, it proved impossible to tag and relocalise the entire pool of PRPF38A protein. Some of the PRPF38A lost its tag and thus could not be “controlled”. This is probably also the reason why the functional effects were not very pronounced.

The manuscript is written very well and is a pleasure to read. The experiments are well controlled and the results, with a few exceptions (see below), are presented appealingly. Certainly, the method is interesting mainly for a limited audience, but it represents a good alternative for the functional inactivation of nuclear proteins that cannot be depleted by other methods.

We share the reviewer's opinion that our work is a proof-of-principle study for splicing factor re-localization as an alternative to other strategies to inactivate or modulate splicing factor function, as we also clearly point out in the manuscript. We thank the reviewer for considering our manuscript as well written and the results as presented in an appealing fashion.

Specific comments:

1. This is probably too much to ask for in such a proof-of-principle study - but it would of course be great if the authors could show the effect of relocalization with another protein. U2AF35 would be an obvious choice here, as it has been shown to be well-behaved in the authors' preliminary experiments. Perhaps the authors have already worked on an endogenously tagged U2AF35 and first results are already available?

We agree with the reviewer that applying the same strategy to endogenously tagged U2AF35 would be interesting. However, as also hinted at by the reviewer, we believe that such an analysis would go beyond the scope of the present study, as it would require time-consuming cell line generation involving CRISPR/Cas knock-in and eventually stable integration of the anchor protein. We agree that there are many potentially interesting further applications, which we will explore in the future.

2. The title is not quite clear. I think it should confer the message that the process of recruitment to the nuclear lamina leads to the inactivation of the splicing factor.

We thank the reviewer for this suggestion. We now changed the title of this manuscript to: **“Recruitment of a splicing factor to the nuclear lamina for its inactivation”**

3. I am not too happy with the way the data are shown in Figure 6.

a. The illustration in Figure 6b can certainly be improved and possibly also include some sort of quantification of the RNA-Seq data.

We now improved the appearance of the sashimi plots (new Figure 6b) and included the RNAseq-based quantification of the targets shown in the sashimi plots of Figure 6b as box plots (new Figure 6c):

We additionally included the following descriptions in the Methods section (line 614):

For sashimi plots (Figure 6b), "junction read coverage min" was set to 3-5 per track. For quantifications, iREAD-derived FPKM values were plotted in GraphPad Prism as box-whisker plots including individual data points (horizontal lines, medians; whiskers, minimum and maximum values).

b. Are the validation PCRs in Figure 6c really the best that the authors can show? For example, I find it hard to believe that the band for *cdc6* in lane 3 can be accurately quantified.

Please note that intron-containing RNAs are of comparatively low abundance and often only constitute a small percentage of the transcripts of the respective gene. Therefore, the intensities of the bands representing intron-retained transcripts in the radioactive gels are comparably low. Still, reliable quantification was possible, as we performed the radioactive PCRs shown in Figure 6 (now Figure 6d) repeatedly with reproducible effects. We include below examples of replicate radioactive gels, which are fully in line with the data we show in the manuscript. We, therefore, kept the original gel images for these experiments in the manuscript, as replicates were of comparable quality.

c. I find it surprising that Figure 6d was normalized to the rCD1 condition. I would have rather set the DMSO control as 1 to show the intron retention as an increase.

We thank the reviewer for the comment and have now normalized the values to the DMSO control. The new Figure 6e was adjusted accordingly:

Reviewer #2 (Remarks to the Author):

Vester et al. elucidate how to mislocalize a splicing factor from its native localization to perturb its function in order to study its effect on mRNA splicing. To this end, they employ reversible chemically induced dimerization. Upon induction, target protein and different anchor proteins are dimerized, which should lead to a change of localization of the target protein to match the localization of the anchor protein. The authors evaluate several possible anchor proteins and show that the best effect is achieved with lamin A, leading to localization of target protein to the nuclear lamina within less than an hour with possible reversion in a few minutes.

This work introduces a useful technology into the field of splicing, which will likely enable many new insights. This is highlighted by novel findings about the effect of mislocalization of splicing factor PRPF38A, which results in changes in intron retention, mostly enhancing intron retention.

The study was conducted in a scientifically sound manner and is well presented in a logical and easy-to-follow fashion. The manuscript is well written.

I strongly appreciate that different difficulties in establishing the system are discussed within the manuscript, which is very beneficial for potential users of this technique.

We thank the reviewer for the very positive overall evaluation of our work, specifically for evaluating the technology as useful, for considering our study to have been conducted in a scientifically sound manner and to be well presented.

There are a few questions and minor issues that should be addressed in my opinion. They are presented here in order of appearance in the text:

1) Please mention the study organism/model in the abstract, to make it easier to grasp.

We now included the cell type used in this study in the abstract (line 26):

In a proof-of-principle study, the partial re-localization of the PRPF38A protein to the nuclear lamina in HEK293T cells induced a moderate increase in intron retention.

Please note that the first sentence of the abstract also mentions “humans” as the organism the study refers to.

2) l. 161 and following: the fact that the mislocalization of PRPF38A also scales with transfection efficiency of the anchor construct (as seen by fluorescence remaining in the nucleoplasm) does not become clear to me within the text, but is mentioned in the figure caption. It would be good to state this more clearly in the text as well.

We thank the reviewer for pointing this out. We added a corresponding explanation to the respective section of the main text (line 206):

For cells with a high relative expression level of the lamin A construct, the recruitment worked efficiently, while for cells with a relatively low expression level of the lamin A construct, the recruitment was incomplete, as indicated by higher red fluorescence remaining in the nucleoplasm (Figure 2d). Therefore, with higher transfection and expression efficiency of the anchor construct, a more efficient recruitment can be achieved.

3) l. 254 and following: The remaining pool of PRPF38A without FKBP is a problem that may prevent seeing more drastic effects. It would be good if the authors could point to how this issue might be improved. E.g., would it be possible to add a second, C-terminal

tag, or to remove the unstructured N-terminus to enable more complete re-localization? In my opinion this does not call for additional experiments but should be discussed.

A removal of the unstructured N-terminus of PRPF38a might result in a non-functional protein, the specific function of this region is presently unknown. However, we thank the reviewer for the suggestion that an (additional) C-terminal tag might present an alternative, possibly giving rise to a fusion protein that will not suffer from protease cleavage (or leading to a larger fraction of PRPF38A maintaining at least one tag). We amended the Discussion accordingly (line 374):

Clearly, the effects observed in our setup are attenuated by the pool of non-recruitable PRPF38A that remains probably due to proteolytic cleavage of the fused FKBP tag within the cells. This problem may be alleviated by fusing a FKBP tag to the PRPF38A C-terminus, which may be less prone to protease cleavage. Alternatively, fusion of FKBP tags to both PRPF38A termini might ensure that all endogenous PRPF38A molecules retain at least one tag *via* which they can be reversibly anchored away.

4) l. 267: The authors state: “To obtain a higher expression of the anchor protein fusion over the target protein fusion, we stably integrated the lamin A-ECFP269 SNAP-coding region into the genome of the C1 CRISPR cell line.”

The stable integration itself only warrants higher expression of anchor protein fusion over target protein if the construct is indeed expressed more strongly than the target. Was this tested, and if so, how? Or was a specific promotor chosen to ensure high expression? This could be stated more clearly.

We expected a higher expression level of the anchor protein due to (a) most likely multiple copies of the lamin A-ECFP-SNAP-coding region becoming integrated upon stable transfection and (b) the strong CMV promoter that we used to drive expression of the construct after integration. We now clarified this rationale in the revised manuscript (line 263):

To obtain a higher expression level of the anchor protein fusion over the target protein fusion, we stably integrated the lamin A-ECFP-SNAP-coding region into the genome of the C1 CRISPR cell line. Stable integration usually results in the integration of several copies of the insert and our vector contained a strong CMV promoter, so that a high expression level of the anchor protein fusion after stable integration into the genome was expected. In contrast, the target FKBP-PRPF38A is present only at endogenous levels after tag knock-in. Stable integration of the anchor lamin A-ECFP-SNAP construct was validated on the DNA level by PCR analysis, revealing the presence of regions encoding ECFP and SNAP (Figure 5a,b).

5) In the discussion, I felt that some points might be missing. Specifically, since the method resembles the Anchor-away strategy (Haruki et al., 2008) that has been used in yeast to achieve re-localization of nuclear factors from the nucleus to other compartments after induced dimerization, I would have expected a mention of this technique. Also, the strategy has recently been adapted to mammalian and other non-yeast cells in a few instances (Galloy et al., 2021, Sanchez Bosch et al., 2020, I may have missed others). I feel that those other approaches should be acknowledged and compared to the method developed here. Might these methods even be orthogonal?

We thank the reviewer for pointing this out and apologize for not having referred to these pioneering studies of these colleagues. We now expanded the Discussion accordingly (line 429):

Apart from the rCD1-based method presented here, other chemical dimerizer-based techniques have been reported for targeted protein re-localization. One of the first chemical dimerizer-based “anchor-away” systems developed was based on rapamycin, which mediates

the dimerization of FKBP12-tagged and FRB-tagged proteins⁸¹, and which was used to recruit nuclear proteins to distinct cellular locations in yeast, e.g., to the cytoplasm *via* tagged RPL13⁸². The same approach was subsequently applied in a proof-of-principle study in *Drosophila melanogaster* to study loss-of-function phenotypes of nuclear proteins⁸³. A similar strategy has been applied in mammalian cells, involving abscisic acid-mediated dimerization of phytohormone factor fusion proteins, with the results suggesting that it could be used to probe the function of chromatin-modifying proteins with reduced side-effects⁸⁴. However, most of these alternative strategies either require extensive washout to reverse the effects or the effects cannot be efficiently reversed due to a high affinity of the chemical dimerizer to the target protein fusion (as, e.g., in the case of rapamycin)⁸⁵. As rCD1-based recruitment can be rapidly reversed, the approach presented here adds additional versatility to the chemical dimerizer-based toolbox for nuclear proteins.

6) In particular, Galloy et al. 2021 used an RPL13 fusion to successfully mislocalize a nuclear protein. I would be curious about possible reasons why RPL13 fusion was less successful in the approach presented here compared to Galloy et al.

We again thank the reviewer for pointing this out. In the first version of the manuscript we had included a short consideration of the mis-localization of tagged RPL13 (line 131):

Although ribosomal proteins are usually localized to the cytoplasm, their over-expression can lead to nucleolar accumulation, because the proteins are in excess over other ribosomal components and, thus, not assembled into ribosomal particles and not exported into the cytoplasm^{39,40}.

We now additionally discuss the RPL13 fusion in more detail (line 444):

In the above-mentioned studies, yeast RPL13A or the *Drosophila*/ human homologs RPL13 were used as anchors to mis-localize nuclear proteins to the cytoplasm, as ribosomal proteins are imported into the nucleus after their production in the cytoplasm, where they are assembled into ribosomal complexes that are subsequently exported again to the cytoplasm. In our study, over-production of RPL13-ECFP-SNAP led to an accumulation of the anchor fusion construct in nucleoli, presumably because it was not efficiently or completely integrated into ribosomal complexes for nuclear export. While CRISPR/Cas9-based SNAP-tagging of endogenous RPL13 may in principle prevent these effects, homozygous tagging of RPL13 could not be achieved in a past study⁸⁴, possibly because of loss of RPL13 functionality in the ribosome.

7) L. 445: Targeted depletion is described as very slow, taking “at least 4-8 hours” to achieve degradation for most cases. However, descriptions of the auxin induced degron system in mammalian cells state rapid depletion with half-lives of 20-40 minutes for the standard AID and around 20 min for AID2 (Yesbolatova et al., 2020). In the latter, it looks like the vast majority of protein will be degraded after 90 minutes. Also, the cited paper by Habet et al. on the dTAG system shows robust degradation to baseline levels of many of the proteins tested within 1 hour (Habet et al., Fig. 4a). Therefore, I believe the authors should reconsider the representation of these methods. A main advantage of the new system presented here clearly is the possibility of very fast recovery of protein localization within few minutes. To me it is not clear whether the 2 papers cited here at the end of the sentence at l. 446 refer to degradation time or recovery time (I agree recovery is much slower in degradation methods). If the authors are aware of particularly slow degradation methods that are widely used, they might want to include citations directly after “at least 4-8 hours” in l. 445.

We again thank the reviewer for alerting us to this point and agree that a more thorough discussion of the degradation systems is required. We amended the Discussion accordingly (line 454):

In addition to RNAi-based strategies, targeted protein degradation offers additional possibilities for loss-of-function analyses. Although the degradation of a target protein can be achieved within hours for many degradation systems (e.g., about 1-8 h for the dTAG system⁹⁰ and about 4-8 h for the HaloPROTAC system⁹¹), fast degradation of some proteins within 20-30 minutes has been described (e.g., using auxin-inducible degradation technologies^{92,93}). The degradation kinetics depend on the target protein subjected to the degradation and can usually not be predicted. Irrespectively, the reversion of the depletion, if possible, is even much slower, in contrast to the method presented here.

8) In the Methods section dilutions of secondary antibodies for the Western Blots are not stated at l. 491-492.

We now included the dilution factors of the secondary antibodies (line 520).

Detection was done with the following antibodies: anti-PRPF38A NBP2-37697 antibody (Novus Biologicals) 1:2000 diluted or NBP2-33673 antibody (Novus Biologicals) diluted to 0.4 µg/ml; anti-actin MAB1501 antibody (Merck Millipore) 1:4000 diluted; anti-GFP sc-9996 antibody (Santa Cruz) 1:1000 diluted; goat anti-rabbit 31460 antibody (Thermo Fisher Scientific) 1:10000 diluted; goat anti-mouse 31430 antibody (Thermo Fisher Scientific) 1:5000 diluted.

9) L. 503: I assume the bands were not detected directly but a storage phosphor screen was exposed to the dried filter paper and then scanned on the Typhoon scanner.

We apologize for the missing part in the description of radioactivity detection and included it now in the respective section (line 536):

The gel was transferred to a filter paper, dried and radioactive bands were detected on a Typhoon phosphorimager (GE Healthcare) via a storage phosphor screen after exposure to the filter paper.

10) L. 576: What are treat_stbl samples?

We apologize for the usage of this unclear term and have now adjusted the description (line 608):

In iREAD, introns were considered differentially spliced with a T-test-derived p-value < 0.001 between the FPKM intron values of triplicate rCD1-treated DE samples and the nine control samples.

11) L. 585: For the T-tests it is not stated what type they were (1- or 2-tailed, homoscedastic or heteroscedastic).

The T-tests were two-tailed and homoscedastic, we now added this information in the statistics description (line 621):

Significance was assessed *via* unpaired, two-sided, homoscedastic T-tests; significance indicators: ns, not significant; *, $p \leq 0.05$; **, $p \leq 0.01$; ***, $p \leq 0.001$. Specific p-values are reported in the respective figure legends.

REVIEWERS' COMMENTS:

Reviewer #1 (Remarks to the Author):

The authors have satisfactorily answered all my criticisms. The manuscript can now be published.

Reviewer #3 (Remarks to the Author):

In the revised manuscript, the authors have carefully answered all questions and concerns raised by both referees. I am completely satisfied with all changes and would therefore recommend the revised manuscript for publication without further changes.